# Preparation and Taste Profiling of the Enzymatic Protein Hydrolysate from a by-Product of Red Snow Crab Processing as a Natural Seasoning Compound

**DOI:** 10.3390/foods11233911

**Published:** 2022-12-04

**Authors:** Ga-Yang Lee, Min-Jeong Jung, Jong-Woong Nam, Ah-Ram Han, Byoung-Mok Kim, Joon-Young Jun

**Affiliations:** 1Department of Food Science and Technology, Tokyo University of Marine Science and Technology, Tokyo 108-8477, Japan; 2Food Convergence Research Division, Korea Food Research Institute, Wanju 55365, Republic of Korea

**Keywords:** red snow crab, processing residues, commercial protease, enzymatic hydrolysate, debittering, electronic tongue, flavoring compound

## Abstract

The red snow crab (*Chionoecetes japonicus*) is the most industrially processed in the Republic of Korea, and the meat is very popular, owing to its savory taste and flavor. Its body meat production comprises a two-step separation to increase meat yield. However, during the secondary separation, broken shell debris is occasionally entrained in the meat products, which is a concern for manufacturers. As the residues from first separation contain 39.9% protein, it can be utilized as an enzymatic protein hydrolysate (FPH) rich in free amino acids (FAAs). A combination of flavourzyme and alcalase (1:1) superiorly hydrolyzed the protein of the residues, and the best hydrolysis condition was suggested at 60 °C for 15 h with fourfold water and 2% enzyme addition, achieving a 57.4% degree of hydrolysis. The EPH was mostly composed of FAAs containing most essential amino acids; however, bitter-tasting amino acids accounted for 46.4% of the FAAs. To reduce the bitter taste, different nonvolatile organic acids were considered as masking agents, and citric and malic acids were effective, though the umami taste is slightly decreased. In conclusion, the crab processing residues can be utilized as an FAA-based natural seasoning compound through enzymatic hydrolysis and organic acid treatment.

## 1. Introduction

Red snow crab (*Chionoecetes japonicus*), belonging to class Malacostraca, is distributed in soft grey mud or sandy floors at a depth of 700–1500 m in the East Sea of the Korean coast [1]. The crab meat is very popular among consumers because of its savory taste and flavor, and a large number of the crabs can be caught stably throughout the year, except during the closed season (from July to August), allowing for their industrial utilization. The crab is mainly processed into cooked picked meats and distributed frozen [2]. In addition, the red snow crab is rich in essential amino acids, including glutamic acid, arginine, and glycine, as well as minerals and omega-3 fatty acids. In particular, the crab contains high amounts of lysine, which is likely deficient in people who eat grain as a staple food [3].

Crab processing generally includes carapace and gill removal after washing, picking intestine, sectioning into two pieces, boiling (or steaming), and cooling [4]. For red snow crab, meat separation comprises dividing into body and leg parts. For the body part, the meat is conventionally separated by a two-step separation to increase meat yield. The meat is primarily separated by compression, and the residues containing soft shells and frame meats are processed in a rotary perforated cylinder to separate the remaining frame meats. However, during the secondary separation, debris from broken shells is occasionally entrained in the meat products, which is a concern for manufacturers, because consumers regard the broken shell debris as a foreign substance.

Meanwhile, the demand for natural seasonings associated with healthy and flavorful compounds has been increasing among consumers who value the relationship between food and health [5]. Recently, the market for convenience food that can be simply cooked at home has grown in the Republic of Korea, owing to the increase in double-income families and single-person households, as well as the consequences of the COVID-19 pandemic. Moreover, the seasoning market has also increased, with total sales of seasoning products in the Republic of Korea reaching approximately USD 132 million in 2021, of which natural seasoning products grew continuously by 2.32% every year for the past 5 years [6].

Enzymatic hydrolysis using commercial proteases is a powerful method for developing extracts rich in flavorful peptides and free amino acids from protein resources [7,8]. Fish or crab processing by-products have been considered into utilization of enzymatic protein hydrolysates (EPHs) for their valorization [9,10] and, in some cases, their physiological benefits, such as antioxidant, immunostimulatory, and antihypertensive activities [11,12]. However, many studies have highlighted the bitter taste of EPHs, which is a weakness when applied to seasoning products [7]. There are several debittering approaches for EPHs, and these can be divided into physicochemical methods (chromatographic separation, Mailard reaction, activated carbon treatment, and use of masking agent, including encapsulation) and biological methods (use of selective enzyme, enzymatic deamidation, and plastein reaction) [13,14]. Among these methods, use of masking agent can be easily applied in industry without decreasing the nutritional value of EPHs [14].

The residues generated from first body meat separation of red snow crabs contain functional food components, including proteins and chitin, that can reproduce the flavor of crabs. Therefore, this study investigated the use of these residues as a natural seasoning compound instead of subjecting them to second body meat separation. In this regard, the production of residues after first body meat separation and their protein content were explored, and a suitable enzyme and its best hydrolysis condition were investigated to prepare EPH from the residues. Furthermore, to reduce the bitter taste of EPH, different organic acids were also considered as masking agents by measuring the taste profiles using an electronic tongue.

## 2. Materials and Methods

### 2.1. Materials and Estimation of Residue Generation

The processing residues of red snow crab (*Chionoecetes japonicus*, males, Figure 1) containing soft shells, frame meats, and slight joint shells were obtained following 1st body muscle separation after boiling at a crab-processing factory (Yangyang Fisheries Co., Yangyang, Republic of Korea). In total, 90 crabs with body weights of approximately 380 ± 60 g were used to estimate the amount of residues generated. First body muscle separation was conducted in three batches of 30 crabs each, and the residues were collected and weighed. To quantify the meat portion in the residues, the frame meats were separated using tweezers and weighed. The raw crabs are processed in crab-processing factory and no ethical issues are involved.

### 2.2. Analysis of Moisture and Crude Protein

The moisture (method 950.46) and crude protein (method 976.05) content of the crushed residues were analyzed according to the AOAC method [15].

### 2.3. Preparation of Enzymatic Protein Hydrolysate

To select a suitable enzyme to hydrolyze the residues, four commercial proteases (flavourzyme^®^ 500 MG, neutrase^®^ 0.8 L, alcalase^®^ 2.4 L or protamex^®^; Megazyme Ltd., Bray, Ireland) were used. The residues were chopped using a home blender, and 50 g of chopped residue (Figure 1) was mixed with 150 mL of deionized water (DW). Thereafter, the mixture was heated at 95 °C for 30 min in a shaking water bath (SI-900R; JeioTech, Daejeon, Republic of Korea) to inhibit bacterial growth and endogenous enzyme activity. After cooling to 60 °C, each of the proteases or a combination of two proteases (1:1) was added to 2% (*w/w*) of the residue weight. The mixture was then hydrolyzed at 60 °C for 5 h with shaking (120 rpm) and heated at 95 °C for 5 min to deactivate the enzyme. After cooling, the EPH was obtained by decompression filtration using filter paper (Advantec No. 5A; Advantec Toyo Kaisha Ltd., Tokyo, Japan). After selecting a suitable enzyme, different parameters affecting degree of hydrolysis (DH), such as water addition ratios (1–5-fold of the residues), enzyme concentration (0–4% of the residues), and reaction time (3–20 h), were separately examined to elicit the best hydrolysis condition. To determine the amino acid nitrogen (AN), aliquots of all EPHs produced under different conditions were used, and the remaining EPHs were concentrated to 40″ Brix (con. EPH) using a rotary evaporator and stored at 4 °C before use for organic acid treatment.

### 2.4. Determination of PH and Brix

The pH was determined using a pH meter (SevenEasy S20K; Metteler Toledo International Inc., Columbus, OH, USA), and the Brix of EPH was determined using a digital pocket refractometer (PAL-3; Atago Co., Ltd., Tokyo, Japan).

### 2.5. Determination of Amino Acid Nitrogen and Hydrolysis Degree

The AN content was determined using the formol titration method [16] with slight modifications. Briefly, 10 mL of EPH or DW (blank) was mixed with 20 mL of DW, and the pH was adjusted to 8.5 with a 0.1 N NaOH standard solution. Afterward, 10 mL of 40% formaldehyde solution was added, mixed well, and titrated with a 0.1 N NaOH standard solution. The AN was calculated and expressed as mg per 100 mL of the sample using the following equation (Equation (1)):AN content (mg 100 mL^−1^) = (A − B) × 0.0014 × 1000/S × 100 (1)
where A and B are the titrated volumes (mL) of the sample (A) and DW (B) used, 0.0014 indicates the molar mass of nitrogen, which is equivalent to 1 mL of NaOH (0.1 N), and S is the sample volume (mL).

The DH was defined as the percent ratio of the total AN amount in the EPH recovered per total nitrogen (TN) amount of residues used, which was calculated according to Equation (2). To calculate the total recovered AN amount from the residues, the total volume of EPH recovered was measured and calculated using Equation (3):DH (%) = A/B × 100 (2)
A (mg) = Equation (1)/100 × V (3)
B (mg) = 2160/100 × S (4)
where A is the total amount of AN (mg) in the EPH recovered, B is the TN amount (mg) of the residues used, V is the total volume (mL) of EPH, and 2160 (mg 100 g^−1^) and S are the TN content of the residues and the weight (g) of the residues used, respectively.

### 2.6. Organic Acid Treatment

To evaluate the effect of nonvolatile organic acids on the taste profile of the EPH, each of food-grade citric, malic, and succinic acids (ES Ingredients, Gunpo, Republic of Korea) were dissolved in DW to a concentration of 30% (*w/v*) as a stock solution. Next, organic acids were separately added to the concentrated EPH until the pH reached 7.0. For comparison, 3 M HCl was used.

### 2.7. Determination of Taste Using an Electronic Tongue

The taste of EPHs treated with different organic acids was determined using an electronic tongue (AstreeⅡ E-tongue; Alpha MOS, Toulouse, France) equipped with a sensor array (AHS, sourness; PKS, sweetness; CTS, saltiness; NMS, umami; ANS, bitterness; SCS, reference; CPS, reference). For assessment using the electronic tongue, the concentrated EPHs treated with different organic acids, including a control (without acid treatment), were diluted to a Brix of 5.0″ (solid matter: 3.5%) with DW, and 20 mL was used. The aquation time was 120 s, and the relative scores between samples ranged from 0 to 12. The experiment was replicated in quintuple. Taste profiling was performed using AlphaSoft v17.

### 2.8. Analysis of Free Amino Acids

Free amino acids in the EPHs with different organic acids adjusted to a Brix of 5.0″ were analyzed using a high-speed amino acid analyzer (L-8800; Hitachi High-Technologies Co., Tokyo, Japan), according to the method described by Kim et al. (2016b) [17]. Briefly, after filtration with a 0.2 µm MCE syringe filter unit, 5 µL of the sample was injected without further treatment and flowed at 0.35 mL min^−1^ with a lithium citrate buffer with ninhydrin reagent. The analysis was performed using an ion exchange resin column (4.6 × 60 mm; Hitachi High-Technologies Co.). The oven temperature was increased from 30 to 70 °C (0.5 °C min^−1^), and the wavelength was measured at 570 and 440 nm (for proline).

### 2.9. Data Analysis

All data, except for free amino acids, were statistically assessed using SPSS (SPSS Inc., Chicago, IL, USA). The values are expressed as the mean ± standard deviation (SD), and a significant difference (*p* < 0.05) in the means was identified by Tukey’s test.

The principal component analysis (PCA) between the type of organic acid and taste profile was conducted using R studio software version 1 July 2022 + 554 (R Foundation for Statistical Computing, Vienna, Austria) containing ‘FactoMineR’ v2.4 and ‘Factoextra’ v1.0.7 packages. PCA data were obtained using the ‘pca’ function of the ‘FactoMineR’ package [18] and visualized using the ‘fvis_pca_biplot’ and ‘fvis_pca_ind’ functions of the ‘factoextra’ package [19].

## 3. Results and Discussion

### 3.1. The Generation of Residues and Their Protein Content

The amounts of residues and their crude protein content are listed in Table 1. The residues accounted for approximately 8.9% of the total body weight of raw crab and contained 68.7% soft shells and 31.3% frame meats. From these data, the maximum amount of the frame meat obtained from the second body meat separation was estimated to be approximately 2.9% (wet basis) of the total body weight. The crude protein content of the residues was 13.5 ± 1.4% (dry basis: 39.9%). The residues include soft shells and frame meats; the meat is mostly composed of protein, and general crustacean shells contain 30–40% protein, 30–50% calcium carbonate, and 20–30% chitin, although species, part, and capture season variations exist [20].

### 3.2. Enzyme Selection for Hydrolysis of the Residues

Enzymatic hydrolysis using commercial proteases is a powerful method for developing extracts rich in flavorful peptides and amino acids from protein resources [7,8]. To select a suitable enzyme for hydrolyzing protein residues, four commercial proteases were used singly or in combination, and the results for the degree of hydrolysis (DH) are shown in Figure 2. The DH of the control without any proteases was determined to be 2.7 ± 0.6%. The DH was highest in the flavourzyme-only treatment (group F, 27.9 ± 2.0%), followed by the alcalase-only treatment (group A, 21.3 ± 1.8%). The DH was significantly increased by a half combination of flavourzyme and alcalase (group ⅟_2_F + ⅟_2_A, 37.0 ± 1.5%) compared with the single treatments (*p* < 0.05). Similarly, it has been reported in some studies that DH or soluble protein amount is increased when enzymes are properly combined rather than when used individually [21,22].

In previous studies, among the industrially available commercial proteases, flavourzyme and alcalase have been observed to be superior in the enzymatic hydrolysis of fish or crustacean proteins. A flavourzyme produced by *Aspergillus oryzae* contains endo- and exo-peptidases and is known to minimize the bitter taste of EPHs [23]. Alcalase produced by *Bacillus licheniformis* acts as an endo-protease that has a high rate of fish protein hydrolysis, although it tends to increase hydrophobic amino acids in EPHs [24].

### 3.3. Elicitation of the Best Hydrolysis Condition

The working ranges of pH and temperature of the four commercial proteases recommended by the manufacturer coincide in the pH range of 6–8 and temperature range of 60–65 °C [25]. The pH of the residues was determined to be approximately 7.9 after mixing with DW; therefore, no pH adjustment was conducted and all reaction temperatures were set to 60 °C. To elicit the best hydrolysis condition for the combined enzyme (flavourzyme-alcalase, 1:1), other parameters that affect the DH of the protein residues, such as water addition ratios, enzyme concentrations, and reaction times, were examined in the order mentioned (Figure 3).

For the water addition ratio, the DHs ranging from onefold to fourfold water additions tended to increase as the water addition ratio increased. However, between fourfold and fivefold water additions (41.8 ± 1.2% and 43.9 ± 3.0%, respectively), there was no significant difference (*p* < 0.05), indicating that fourfold water addition is appropriate because, as water addition ratio increases, the cost for concentration also increases (Figure 3A). Regarding enzyme concentrations, the DH at 2% of the combined enzyme did not significantly differ compared with that at 4% of the combined enzyme (*p* < 0.05), though the DH showed a stepwise increase with increasing enzyme concentrations (Figure 3B). During the reaction, the DH increased until 15 h and was maintained (Figure 3C). In short, the best hydrolysis condition for the protein of the residues was suggested at 60 °C for 15 h with fourfold water addition and 2% of the combined enzyme, achieving a 57.4% DH.

It is difficult to directly compare the DH with other studies because there are differences in substrates and experimental conditions. In addition, there is no case of use in combination with crab shells and meats for enzymatic hydrolysis, as previous studies have been conducted separately; Jang et al. [23] studied the optimization of enzymatic hydrolysis of the meat scraps from red snow crab using flavourzyme to utilize as a flavoring compound; they suggested that the optimal hydrolysis condition was at pH 7.2 and 51.8 °C for 4 h 45 min with 3.8% of the enzyme, resulting in a 58.0% DH. Noh et al. [22] reported that a combination of protease A and either protamex or flavourzyme was suitable for hydrolysis of the leg shell from red snow crab, and the hydrolysis time required up to 12 h.

### 3.4. Organic Acid Treatment and Electronic Tongue-Based Taste Profile

Enzymatic hydrolysis increases the usability of protein resources while maintaining their intrinsic nutritional value [26] and, in some cases, it exerts physiological benefits, such as antioxidant, immunostimulatory, and antihypertensive activities [11,12]. However, the bitter taste associated with the EPH, which is a weakness when applied to food condiments, has been highlighted in many studies [7].

In the present study, the EPH was concentrated to a Brix of 40.0″ to reduce deterioration during refrigeration. Since the pH of the concentrated EPH was 8.2, it was neutralized (7.0) using four types of nonvolatile organic acids that were evaluated as bitter taste masking agents. All EPHs were diluted with DW to a Brix of 5.0″ (dry matter: approximately 3.6 g 100 mL^−1^) before measurement by the electronic tongue. The taste profiles were represented by five sensory attributes (umami, saltiness, sweetness, sourness, and bitterness) (Figure 4A). HCl was used as a comparative acid to understand the effect of neutralization. The control without any acids exhibited the highest scores for bitterness, umami, and saltiness, whereas the bitterness and saltiness were notably decreased by treatment with citric and malic acids and HCl. The sweetness and sourness of all organic-acid-treated residues increased but, in the case of HCl, the sweetness did not increase.

Figure 4B shows two-dimensional scatter plots statistically assessed by principal component analysis (PCA) between the types of organic acids and taste profiles. The dimensions (Dim) 1 and 2 on the PCA plot explained 55.2 and 31.1% of the total variance, respectively. Umami and saltiness were positively loaded on Dim 1 and were inversely related to sweetness and sourness; the organic acid treatments influenced these relationships. Dim 2 explained the variances of bitterness (positively loaded) and sourness (negatively loaded), indicating that adding acids to the EPH reduced bitterness. According to the qualifying classification, there were similarities between all EPHs with organic acids that were loaded on opposite sides of the control, and those with HCl had other directions.

### 3.5. Free Amino Acids

The occurrence of bitter taste has been found in various fish protein hydrolysates [24,27]; EPHs are primarily composed of peptides and amino acids, the main causative components of bitter taste. There are two possibilities regarding the manifestation of bitter taste in EPHs: one is the formation of specific proteolytic peptides in the hydrophobic amino acids located in the C-terminal [13,28]. The other is the accumulation of bitter-tasting amino acids that have been classified in L-forms of valine, leucine, isoleucine, methionine, phenylalanine, tryptophan, histidine, tyrosine, lysine, and arginine [24,29]. The free amino acid compositions of EPHs treated with different organic acids (Brix 5.0″; dry matter: approximately 3.6 g 100 mL^−1^) are listed in Table 2. The total free amino acid (FAA) content of the control without any acids was determined to be 1743.8 mg 100 mL^−1^, indicating that FAAs comprised 48.4% of the dry matter. Leucine, lysine, alanine, glutamic acid, ornithine, glycine, phenylalanine, valine, and isoleucine were the major amino acids, and all the essential amino acids, except for threonine and tryptophan, had a total content of 745.7 mg mL^−1^ in the control. The bitter-tasting amino acids in the control were determined to be 809.8 mg 100 mL^−1^, accounting for 46.4% of the FAAs. With organic acid treatment, there were no notable changes in the FAA composition, though phosphoserine and ammonia decreased and threonine slightly increased, regardless of acid type.

Generally, there are two ways to mask the bitter taste of amino acids: one is to physically prevent bitter molecules, e.g., by encapsulation, and the other is the use of additives, such as salt, sweeteners, flavors, or organic acids, to confuse the brain [30]. The former is expensive and unsuitable for producing a seasoning product, and using salt and sweeteners poses potential health issues, such as hypertension, diabetes, and obesity [31]. On the other hand, organic acids play a key role in providing freshness and sweetness in small amounts in fish sauces, which are rich in peptides and amino acids [17]. In the present study, although the umami taste of EPH was slightly decreased with citric and malic acid treatment, even in small amounts capable of neutralizing the pH, it was effective in masking the bitter taste of EPH.

## 4. Conclusions

This study was conducted to consider utilization of the residues from first body meat separation as a natural seasoning compound. A suitable enzyme and its best hydrolysis conditions were investigated to prepare an EPH from these residues. A combination of flavourzyme and alcalase (1:1) superiorly hydrolyzed the protein of the residues, and the best hydrolysis condition was suggested at 60 °C for 15 h with fourfold water and 2% enzyme addition, achieving a 57.4% degree of hydrolysis. The EPH was mostly composed of FAAs, containing most essential amino acids. However, bitter-tasting amino acids accounted for 46.4% of the FAA. To reduce the bitter taste of EPH, different nonvolatile organic acids were tested as masking agents, and citric and malic acids effectively reduced the bitter taste, with a minimal decrease in the umami taste. In conclusion, the crab processing residues can be utilized as an FAA-based natural seasoning compound through enzymatic hydrolysis and organic acid treatment, and this might be a good way for upcycling and valorization of the crab processing residues.

## Figures and Tables

**Figure 1 foods-11-03911-f001:**
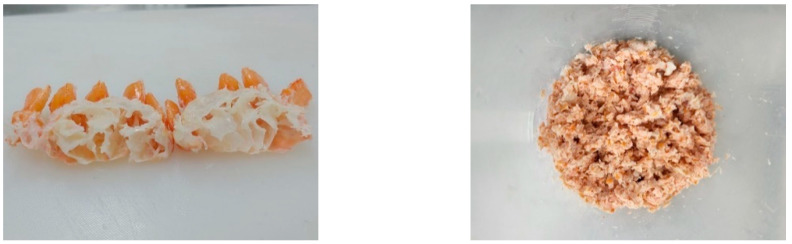
Images of the residues obtained from 1st body meat separation (**left**) and a crushed sample (**right**).

**Figure 2 foods-11-03911-f002:**
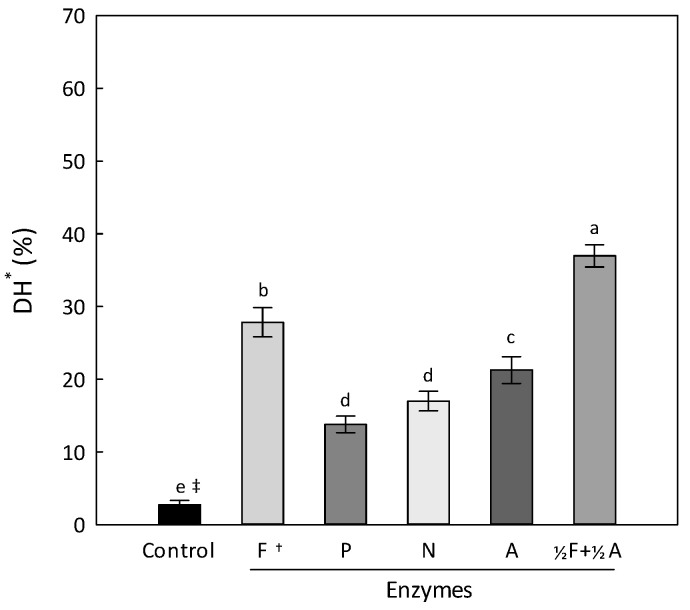
The degree of hydrolysis of the protein in the residues using different commercial proteases. Data are expressed as the mean ± SD of three samples. * Degree of hydrolysis. Control was treated without any enzyme. ^†^ F, flavourzyme 2%; P, protamex 2%; N, neutrase 2%; A, alcalase 2%; ⅟_2_F + ⅟_2_A, flavourzyme 1% plus alcalase 1%. ^‡^ Different letters indicate significantly different values (*p* < 0.05). The enzymatic hydrolysis was conducted under the following conditions: pH, 7.9; enzyme concentration, 2% (*w/w*) of residue weight; ratio of water addition, 3-fold of the residues; reaction temperature and time, 60 °C and 5 h, respectively.

**Figure 3 foods-11-03911-f003:**
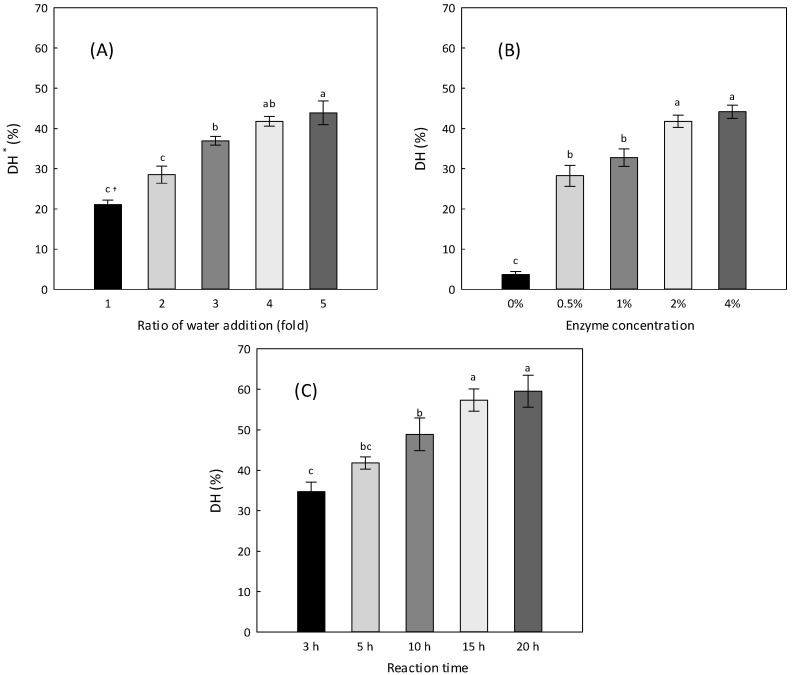
The change in the degree of hydrolysis (DH) of the protein in the residues according to the ratio of water addition (**A**), enzyme concentration (**B**), and reaction time (**C**). Data are expressed as the mean ± SD of three samples. * Degree of hydrolysis. ^†^ Different letters indicate significantly different values (*p* < 0.05). The pH and reaction temperature for enzymatic hydrolysis were fixed at 7.9 and 60 °C, respectively, and other parameters were set separately as follows: (**A**) enzyme concentration, F + A (1:1) 2% (*w/w*) of residue weight; reaction time, 5 h; (**B**) ratio of water addition, 4-fold of the residues; reaction time, 5 h; (**C**) ratio of water addition, 4-fold of the residues; enzyme concentration, F + A (1:1) 2% (*w/w*) of residue weight.

**Figure 4 foods-11-03911-f004:**
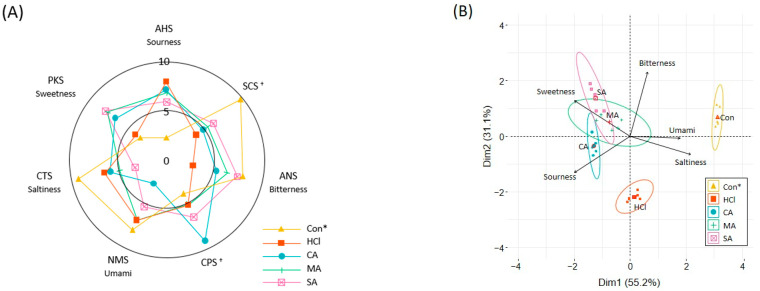
The effect of organic acid on the taste profile of enzymatic protein hydrolysate (**A**) and the relationship between organic acid and taste profile by principal component analysis (**B**). Data are expressed as the mean ± SD of five samples. * Con, without any acids; HCl, hydrochloric acid; CA, citric acid; MA, malic acid; SA, succinic acid. ^†^ Reference sensors.

**Table 1 foods-11-03911-t001:** The amount of the residues obtained from 1st body muscle separation and their crude protein content.

Proportion in the Total Body Weight (g 100 g^−1^ a Raw Crab, Wet)	Moisture Content (g 100 g^−1^)	Crude Protein Content (g 100 g^−1^, Wet)	Weight Ratio of Shells vs. Meats (%, Wet)
8.9 ± 1.0	66.2 ± 4.8	13.5 ± 1.4	68.7:31.3

The data, except for the weight ratio of shells to meats, are expressed as mean ± SD in triplicate. The average total body weight of the raw crabs was 380 ± 60 g.

**Table 2 foods-11-03911-t002:** Differences in the free amino acids of the enzymatic protein hydrolysates according to organic acid treatment.

Amino Acid	Control		Hydrochloric Acid		Citric Acid		Malic Acid		Succinic Acid	
	mg 100 mL^−1^	(%) *	mg 100 mL^−1^	(%)	mg 100 mL^−1^	(%)	mg 100 mL^−1^	(%)	mg 100 mL^−1^	(%)
Phosphoserine	25.1	(1.4)	13.2	(0.7)	13.1	(0.8)	15.3	(0.8)	16.3	(0.9)
Taurine	37.3	(2.1)	36.7	(2.1)	35.6	(2.1)	37.7	(2.1)	37.6	(2.1)
Aspartic acid	22.6	(1.3)	25.3	(1.4)	23.1	(1.3)	22.8	(1.3)	22.0	(1.2)
Threonine	65.6	(3.8)	78.7	(4.4)	76.6	(4.4)	79.6	(4.4)	78.2	(4.3)
Serine	46.6	(2.7)	48.3	(2.7)	46.8	(2.7)	49.3	(2.7)	49.1	(2.7)
Glutamic acid	128.6	(7.4)	137.6	(7.8)	133.1	(7.7)	140.3	(7.7)	139.3	(7.7)
Sarcosine	14.1	(0.8)	14.2	(0.8)	13.6	(0.8)	14.2	(0.8)	14.2	(0.8)
α-aminoadionic acid	24.5	(1.4)	25.3	(1.4)	24.6	(1.4)	26.0	(1.4)	25.9	(1.4)
Glycine	123.7	(7.1)	124.8	(7.0)	121.3	(7.0)	128.1	(7.0)	127.6	(7.1)
Alanine	143.6	(8.2)	142.3	(8.0)	138.8	(8.0)	146.6	(8.1)	146.4	(8.1)
Citrulline	61.6	(3.5)	62.3	(3.5)	60.5	(3.5)	64.1	(3.5)	63.7	(3.5)
α-aminobutyric acid	13.1	(0.7)	13.4	(0.8)	13.0	(0.8)	13.8	(0.8)	13.7	(0.8)
Valine	115.6	(6.6)	115.9	(6.5)	113.0	(6.5)	119.3	(6.6)	119.0	(6.6)
Cysteine	22.3	(1.3)	25.7	(1.4)	24.9	(1.4)	26.3	(1.4)	25.7	(1.4)
Methionine	60.5	(3.5)	61.6	(3.5)	60.2	(3.5)	63.5	(3.5)	62.7	(3.5)
Isoleucine	94.8	(5.4)	95.0	(5.4)	92.7	(5.4)	98.1	(5.4)	97.0	(5.4)
Leucine	161.1	(9.2)	162.4	(9.1)	158.6	(9.2)	167.6	(9.2)	165.4	(9.2)
Tyrosine	59.5	(3.4)	62.7	(3.5)	59.9	(3.5)	60.3	(3.3)	60.0	(3.3)
Phenylalanine	117.7	(6.7)	117.2	(6.6)	114.4	(6.6)	121.0	(6.7)	118.6	(6.6)
β-alanine	5.3	(0.3)	5.2	(0.3)	5.1	(0.3)	5.3	(0.3)	5.0	(0.3)
β-aminoisobutyric acid	10.6	(0.6)	10.8	(0.6)	10.6	(0.6)	11.2	(0.6)	10.5	(0.6)
γ-aminobutyric acid	2.5	(0.1)	2.4	(0.1)	2.4	(0.1)	2.5	(0.1)	2.3	(0.1)
Ammonia	1.4	(0.1)	0.7	(≤0.1)	0.7	(≤0.1)	0.7	(≤0.1)	0.8	(≤0.1)
Hydroxylysine	2.5	(0.1)	2.5	(0.1)	2.4	(0.1)	2.5	(0.1)	2.5	(0.1)
Ornithine	128.9	(7.4)	132.2	(7.4)	128.8	(7.5)	136.1	(7.5)	135.6	(7.5)
Lysine	154.2	(8.8)	155.3	(8.7)	151.5	(8.8)	160.0	(8.8)	159.4	(8.8)
Histidine	41.9	(2.4)	43.6	(2.5)	42.4	(2.5)	44.9	(2.5)	44.6	(2.5)
3-methylhistidine	4.3	(0.2)	3.4	(0.2)	3.3	(0.2)	3.5	(0.2)	3.5	(0.2)
Anserine	19.6	(1.1)	20.5	(1.2)	20.1	(1.2)	21.1	(1.2)	21.1	(1.2)
Carnosine	3.0	(0.2)	3.1	(0.2)	3.0	(0.2)	3.2	(0.2)	3.1	(0.2)
Arginine	4.6	(0.3)	4.8	(0.3)	4.8	(0.3)	5.1	(0.3)	5.0	(0.3)
Proline	27.6	(1.6)	28.3	(1.6)	27.3	(1.6)	28.6	(1.6)	28.5	(1.6)
EAA ^†^	745.7	(42.8)	751.0	(42.3)	732.9	(42.5)	774.4	(42.6)	766.8	(42.5)
BTAA ^‡^	809.8	(46.4)	818.6	(46.1)	797.7	(46.2)	839.8	(46.2)	831.8	(46.1)
Total	1743.8	(100.0)	1775.4	(100.0)	1726.2	(100.0)	1818.8	(100.0)	1804.8	(100.0)

Data are expressed as the mean of duplicate measurements. * Ratio of total free amino acids. ^†^ Essential amino acid. ^‡^ Bitter-tasting amino acids4.

## Data Availability

Data presented in this study are available in the article.

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
