# Peer review of "Preparation and Taste Profiling of the Enzymatic Protein Hydrolysate from a by-Product of Red Snow Crab Processing as a Natural Seasoning Compound"

_foods, 2022, doi:10.3390/foods11233911_

Round 1

Reviewer 1 Report

Totally, the research has been well designed. Although, there are some points to review.

Line 20: Please define EPH in the abstract for the first time

Line 176: 2nd should be changed to second or 2nd.

Figures 2: What is the symbol on the control column? And also in figure 3a on the Control column? Please define 

Please use more relevant new articles, which published by MDPI about peptides and hydrolysate of fish byproducts to compare and improve the discussion of the manuscript. 

Author Response

Response to Reviewer 1,

We are grateful to the reviewer for the valuable comment. We believe your comment will greatly improve the quality of our manuscript. As indicated in the response, we have taken all the suggestions into consideration in the revised manuscript.

We prepared the detail responses to the comments in a separate file.

Reviewer 2 Report

The article under review is devoted to the development of technological regimes for obtaining protein hydrolysates from the waste of primary processing of red snow crab. Enzymatic hydrolysis was chosen as the hydrolysis method. The authors determined the effectiveness of four different enzymes. The influence of technological parameters - hydromodulus, time and concentration of the enzyme on the degree of hydrolysis was studied. The obtained hydrolysate is supposed to be used as a natural seasoning. The topic of work related to the rational processing of bioresources is certainly relevant.

When reading the manuscript, several remarks arise.

1. Abstract. In the abstract, it is necessary to formulate the main idea of ​​the research for the reader. it is necessary to give an explanation of the used abbreviation EPH.

2. Introduction. In the introduction, there are practically no literature data on the use of enzymes for the hydrolysis of protein-containing raw materials, for example, crab processing waste, fish processing waste). There are no literature data on the technological regimes of enzymatic hydrolysis.

3. Results and discussions. What degree of hydrolysis do the authors consider sufficient (optimal) for the studied hydrolysates? Why did the authors settle on the degree of hydrolysis of 57%?

4. Lines 232-234: "the best residue hydrolysis occurred with 4-fold water addition and 2% of the combined enzyme at 60 °C for 15 h". Why was this choice made? As Figure 3 shows, at a hydromodulus of 1:5 (Fig. 3a), at an enzyme concentration of 4% (Fig. 3b) and at a duration of 30 h (Fig. 3c), the degree of hydrolysis is higher.

5. Line 230: "..citric and malic acids effectively reduced the bitter taste..".  Figure 4 shows the reduction in bitter taste with these acids. At the same time, the data in Table 2 show that the content of Bitter-tasting amino acids does not decrease when the hydrolysate is treated with citric and malic acids compared to other acids. Need to provide explanations.

Author Response

Response to Reviewer 2,

We are grateful to the reviewer for the valuable comment. We believe your comment will greatly improve the quality of our manuscript. As indicated in the response, we have taken all the suggestions into consideration in the revised manuscript.

We prepared the detail responses to the comments in a separate file.

Reviewer 3 Report

I am very grateful you for the invitation to review the manuscript foods-2030408 by Lee and coauthors "Preparation and taste profiling of the enzymatic protein hydrolysate from a by-product of red snow crab processing as a natural seasoning compound”. This study investigated the use of these residues as natural seasoning instead of subjecting them to 2nd body meat separation. In this regard, the production of residues after 1st body meat separation and their protein content were explored, and a suitable enzyme and its optimal hydrolysis conditions were investigated to prepare EPH from the residues. Furthermore, to reduce the bitter taste of EPH, different organic acids were also considered as masking agents by measuring the taste profiles using an electronic tongue. The work is very interesting but needs adjustments to increase the quality of the material.

Comments:

- Abstract, Line 16-17: The sentence does not make clear the production of residues and the possibility of transformation. Please review the sentence.

- Abstract, Line 16: Please indicate the average concentration of proteins in the residue.

- Abstract, Line 17: The application as “natural seasoning compound” is not clear in the sentence.

- Abstract, Line 20: Completely specify the EPH acronym on the first appearance. The abbreviation appears only in the Introduction.

- Abstract, Lines 24-25: A completer and more targeted conclusion should be added.

- Abstract: Briefly include the steps for obtaining the natural seasoning compound (methodology steps).

- Lines 26-27: Change the repeated keywords by different words from the title.

- Lines 44-46: Please indicate conditions that influence debris carryover to meat and whether this can be minimized.

- Lines 44-46: It is unclear whether this makes production unfeasible or alters sensory properties. Review.

- Introduction: Include information about the red snow crab market and consumption.

- 2. Materials and Methods: Indicate on which samples the analyzes were performed (crude, treated, or both).

- Line 112: Include the abbreviation "AN" before its use.

- 2.6. Organic acid treatment: It is not clear the treatment, objective and the absence of methodological reference is noted.

- Line 151: Review how references are presented.

-  Lines 169-173: Please avoid repetition of sentences already presented.

- Figures 2 and 3: Consider modifying the DH axis up to 60%.

-  3.3. Elicitation of the optimal hydrolysis condition: It is not clear if the parameters were fixed for the later test or if they all started from the same condition, but using the enzyme, time, and dilution variations, for example.

Author Response

Response to Reviewer 3

We are grateful to the reviewer for the valuable comment. We believe your comment will greatly improve the quality of our manuscript. As indicated in the response, we have taken all the suggestions into consideration in the revised manuscript.

We prepared the detail responses to the comments in a separate file.

Round 2

Reviewer 2 Report

The authors took into account the reviewer's comments and made the necessary additions and corrections to the manuscript. The article can be recommended for publication.

Author Response

Dear Reviewer,

We are grateful to reviewer's comment.